# BEYOND FIXED RESOLUTION: ENHANCING VLLMs WITH ADAPTIVE INPUT SCALING

## ABSTRACT

Real-world vision-language applications demand varying levels of perceptual granularity. However, most existing visual large language models (VLLMs), such as LLaVA, pre-assume a fixed resolution for downstream tasks, which leads to subpar performance. To address this problem, we *first* conduct a comprehensive and pioneering investigation into the resolution preferences of different vision-language tasks, revealing a correlation between resolution preferences with ❶ image complexity, and ❷ uncertainty variance of the VLLM at different image input resolutions. Building on this insight, we propose an empirical formula to determine the optimal resolution for a given vision-language task, accounting for these two factors as the zeroth-order and first-order terms in the Taylor expansion on a given image input. *Second*, based on rigorous experiments, we propose a novel parameter-efficient fine-tuning technique to extend the visual input resolution of pre-trained VLLMs to the identified optimal resolution. Extensive experiments on various vision-language tasks validate the effectiveness of our method.

## 1 INTRODUCTION

Visual Large Language Models (VLLMs) represent a powerful class of models capable of handling vision-language tasks (Yin et al., 2023; Liu et al., 2023a; 2024; Alayrac et al., 2022). There is a growing body of research focused on the application of VLLMs in real-world scenarios, where different tasks necessitate varying levels of perceptual granularity. For instance, autonomous driving systems require high resolution to capture multiple objects and intricate details (Zhou et al., 2023; Ding et al., 2023), whereas image classification tasks involving singular, simple objects can be effectively performed at lower resolutions (Li et al., 2024a; 2023d; Zhang et al., 2024). Despite this, most existing VLLMs, *e.g.*, LLaVA, pre-assume a fixed resolution for downstream tasks, which leads to sub-optimal performance (Liu et al., 2023b;a; Li et al., 2023b). A direct "*exhaustive training*" strategy to adapt current VLLMs for diverse vision-language applications by training the models at different resolutions during the pre-training phase to create a series of checkpoints corresponding to various image input resolutions, followed by the selection of the most effective checkpoint for downstream tasks. While this method is viable, it incurs significant training costs. Consequently, we pose the first research question (***RQ1***):

> ***RQ1***: *For a given vision-language task, how to accurately determine the optimal resolution **without such exhaustive training** for VLLMs?*

To answer ***RQ1***, we conduct a comprehensive and pioneering investigation into the resolution preferences across eight widely-studied vision-language tasks, utilizing VLLMs with five varying input image resolutions, as shown in Figure 1. Our findings reveal that directly choosing the lowest ($224^2$) and highest ($672^2$) resolution leads to subpar performance across tasks. On the other hand, we observe diverse preferences for the intermediate resolutions, with optimal choices scattered among $336^2$, $448^2$, and $560^2$.

To determine the resolution preference for different tasks, we propose two heuristic methods: ❶ image complexity,

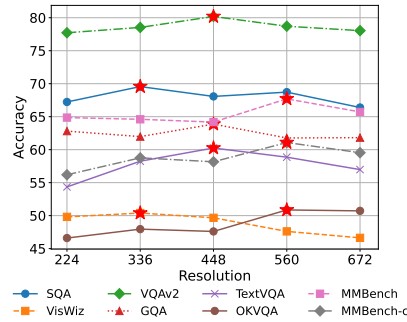

Figure 1: Resolution preference across eight tasks; ★ marks the optimal resolutions.

which measures the intrinsic complexity of a given image, as introduced in Secion 3.2.1. ❷ uncertainty variance, which measures the variance of uncertainty in the model predictions at different image input resolutions, as introduced in Secion 3.2.2. These two heuristic methods can be regarded as the zeroth-order and the first-order terms in the Taylor expansion over image inputs, as discussed in Section 3.2.3. Through empirical analysis across eight vision-language tasks, we find that both the complexity scores and model uncertainty variance exhibit a generally positive correlation with the preferred resolution for each task. Building on this insight, we propose an empirical formula integrating both heuristics to determine the optimal resolution for each vision-language task. We utilize three reference tasks to optimize a single hyperparameter of this empirical formula, and the fitting results across five additional tasks affirm its generalizability.

Once the optimal resolution for a given vision-language task is identified, the next step is adapting the current VLLM to the identified resolution. While the training-free method exists for resolution extension, we empirically find it would lead to performance degradation, suggesting that training-based approaches are essential. However, re-training a VLLM with another resolution from scratch incurs significant costs. This prompts our second research question (**RQ2**):

> **RQ2**: *How to **efficiently** adapt a pre-trained VLLM to the designated resolution without compromise on the performance?*

To tackle this problem, we propose a post-training strategy that extends the image input resolution of an existing VLLM checkpoint. We conduct a preliminary experiment to identify which parameters within the VLLM are crucial for performance enhancement. Based on the findings, we propose a parameter-efficient fine-tuning (PEFT) approach, which only requires updating a few parameters in each VLLM component: the positional embedding parameters of the visual encoder, the projector parameters, and the LoRA adapter parameters of the LLM backbone. Empirical studies show that our method achieves the best efficiency-performance *Pareto front*.

In summary, this paper has the following contributions:

- **Novel Discovery.** Through a comprehensive and pioneering investigation, we discover that different vision-language tasks prefer distinct resolutions.
- **Empirical Formula.** We find these preferences correlated with image complexity and model uncertainty variance on samples at different input image resolutions, which can be interpreted as two terms in a Taylor Expansion of image inputs. We then propose an empirical formula to adaptively determine the optimal resolution for various downstream vision-language tasks without exhaustively training VLLMs.
- **Efficient Adaptation.** We introduce a PEFT approach to extend the input image resolution of LLaVA through post-training, containing three components, including vision module PEFT, language module PEFT, and the projector tuning.

## 2 RELATED WORKS

**Vision Large Language Models.**  Vision Large Language Models (VLLM), as one the most capable and popular solutions to multimodal tasks, extends the reasoning and generating ability of Large Language Model (LLM) beyond language modalities such as image, video, and audio (Alayrac et al., 2022; Liu et al., 2023a; McKinzie et al., 2024a; Tong et al., 2024; Xue et al., 2024). VLLMs can be divided into encoder-decoder and decoder-only VLLM according to their architecture (Liu et al., 2023b; Driess et al., 2023; fuy; Team, 2024). The encoder-decoder VLLM introduces additional multimodal encoders and a modality connector to project multimodal features into the spaces of language models. The implementations of modality connector include: the projector that directly maps features into language model (Liu et al., 2024; 2023a;b); the resampler that compresses the visual feature and inserts cross-gated attention layers into the LLM decoder (Alayrac et al., 2022; Awadalla et al., 2023; Li et al., 2023a). This study mainly focuses on the LLaVA-style VLLM, which adopts encoder-decoder architecture with a projector connector.

**High resolution VLLM**  The high-resolution problem of VLLM is attracting attention because of its prevalence in downstream tasks, such as OCR and document analysis. However, it remains challenging because high-resolution images are underrepresented in the training data, making it difficult to generalize for popular MLLMs. High-resolution VLLM solutions can be roughly divided into two

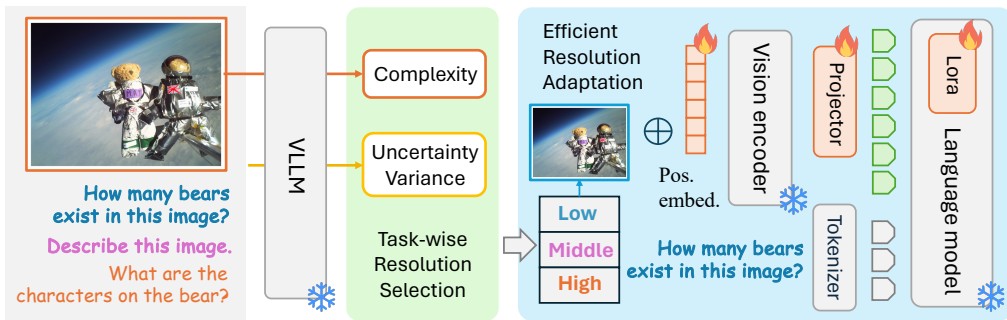

Figure 2: Our method comprises two components: the first component identifies the optimal image input resolution for a given vision-language task (depicted in green), while the second component adapts the VLLM to the selected image input resolution (depicted in blue).

classes: (1) using high-resolution vision encoders that directly support high-resolution input (Hong et al., 2023; Li et al., 2024b; Lv et al., 2023; Wei et al., 2023); (2) the patchification that cuts the high-resolution image into smaller patches to be processed on standard vision encoders (wen Dong et al., 2024; Hu et al., 2024; Feng et al., 2023; Li et al., 2023d; Xu et al., 2024). However, these solutions lack the flexibility for different resolution inputs, which can be computationally expensive. To solve this, FlexAttention uses dual tokenization that only processes a few highly-attended high-resolution tokens in the deeper LLM layers, achieving near 40% reduction in computational cost compared to standard LLaVA (Li et al., 2024a). NVLM (Dai et al., 2024) introduces 1-D tile-tagging for tile-based dynamic high-resolution images, which can significantly improve the performance of OCR-related tasks, but sometimes undermine the accuracy of reasoning-related tasks. Unlike these methods, which presuppose a fixed resolution for downstream applications, our approach implements a task-wise resolution adaptation strategy, employing different resolutions for tasks with different perceptual demands. Additionally, we enhance image input resolution through a parameter-efficient post-training method, circumventing the need for training from scratch to minimize costs.

## 3 METHODOLOGY

This section elaborates on our proposed methodology. Section 3.1 presents an overview, followed by a detailed explanation of each component in Sections 3.2 and 3.3.

### 3.1 METHOD FRAMEWORK

Figure 2 illustrates our approach, which consists of two key components.

The first component focuses on task-specific resolution selection, where we introduce two heuristic approaches to determine the optimal resolution for a given vision-language task, detailed in Section 3.2.1 and 3.2.2. We explore the theoretical connection between these heuristics and the Taylor expansion on image input in Secion 3.2.3, leading to an empirical formula that facilitates task-wise resolution selection in Section 3.2.4.

After identifying the optimal resolution, the second component adapts the VLLM to this specific resolution using a PEFT approach. This involves post-training a existing VLLM checkpoint without retraining the model from scratch. The PEFT adaptation process is discussed in detail in Section 3.3.

### 3.2 TASK-WISE OPTIMAL RESOLUTION SELECTION

As highlighted in Section 1, different vision-language tasks have varying requirements for the perceptual capacity of VLLMs. Therefore, it is critical to do task-wise resolution selection. While tuning VLLMs at different image input resolutions and obtaining the best-performing one is feasible, it imposes heavy training costs, which leads to **RQ1**. In this section, we propose a training-free

method for determining the optimal resolution for a specific vision-language task, utilizing two heuristic approaches.

The first heuristic estimates the complexity of the images for each task, while the second evaluates the variance in model uncertainty at different input resolutions. We then derive an empirical formula to guide the resolution selection process.

### 3.2.1 Measuring Image Complexity

The initial step in VLLM processing is the perception of visual input. Intuitively, images with varying complexity levels demand different degrees of perceptual capacity, with more complex images requiring finer granularity in perception. Thus, for any given vision-language task, image complexity can serve as an indicator of resolution preference.

We use an existing tool (Mahon & Lukasiewicz, 2023) to measure image complexity, which applies hierarchical clustering on image pixels and leverages the minimum description length principle to determine the number of clusters. The average image complexity across samples of the specific task serves as an indicator for determining the appropriate resolution.

### 3.2.2 Measuring Uncertainty Variance across resolutions

In addition to the image complexity, which addresses only the visual aspects of a task, it is crucial to account for the model uncertainty of VLLMs, as it provides insights into the interaction between the visual and linguistic components of vision-language tasks. Furthermore, the method in Section 3.2.1 only captures static complexity, neglecting the effects of varying image resolutions. To complement this, we introduce the second heuristic approach based on model uncertainty.

Specifically, for a VLLM pre-trained at a fixed resolution (e.g., $336^2$ for LLaVA), we extend the visual encoder's resolution using position embedding interpolation, following methods employed in previous studies (Bai et al., 2023; Li et al., 2023b). We denote the original model as $M1$ and the extended-resolution model as $M2$. We first apply random augmentation to the images from the task, following the existing RandAugment algorithm (Cubuk et al., 2020). After augmentation, inference is conducted on the task samples using models $M1$ and $M2$, from which we extract the softmax probabilities corresponding to each generated token. To quantify the uncertainty associated with each token, we calculate the information entropy using $H(p) = -\sum_{i=1}^{n} p_i \log p_i$. Here, $H(x)$ represents the entropy for token $x$, where $p(x_i)$ is the softmax probability of the $i^{th}$ token and $n$ is the number of possible tokens in the vocabulary. We denote the entropy values derived from $M1$ and $M2$ as $U1$ and $U2$, respectively, which provide a measure of uncertainty in the model's predictions.

The uncertainty variance is computed as the ratio of the difference between $U1$ and $U2$ to $U1$, as shown in $r = \frac{U_2 - U_1}{U_1}$. Here, $V(T)$ represents the uncertainty variance for task $T$. This ratio quantifies how much the uncertainty changes between the two VLLMs, with higher values indicating a greater impact of resolution on the model's uncertainty. This ratio is averaged across all generated tokens for a given sample, and the final uncertainty variance is computed by averaging this ratio across all samples in the task.

This heuristic approach serves two functions: (1) it computes entropy based on the tokens generated by VLLM, thus accounting for both visual and linguistic features during inference; and (2) it quantifies the variance caused by resolution changes, thereby capturing the dynamic effects of resolution shifts. Unlike the static image complexity heuristic, this method emphasizes the impact of resolution modifications, making these two heuristics complementary.

Notably, we extend the image input resolution of VLLM without tuning the model parameters, allowing us to avoid additional training costs.

### 3.2.3 Designing Heuristic from the Taylor Expansion Perspective

We further interpret the two heuristics using a Taylor expansion perspective on image inputs. As shown in Equation 1, the Taylor expansion is defined over image inputs, where $I$ represents an image, $R$ denotes its resolution, and $F(I, R)$ represents the overall model evaluation. $C(I)$ denotes the image complexity, which captures the intrinsic properties of the image, while $V(I)$ represents the uncertainty variance, indicating the model's sensitivity to changes in resolution. $\Delta R$ refers to the

difference between two resolutions, and $H(I, R)$ represents higher-order terms related to resolution changes.

$$F(I, R) = C(I) + V(I) \cdot \Delta R + \frac{H(I, R)}{2!}(\Delta R)^2 + \cdots \tag{1}$$

In the simplified form shown in Equation 2, only the zeroth-order and the first-order terms are considered:

$$F(I, R) \approx C(I) + V(I) \cdot \Delta R \tag{2}$$

This simplified expression highlights the inherent complexity of the image ($C(I)$) and the linear change in model uncertainty due to resolution variations ($V(I) \cdot \Delta R$). This framework underlines the importance of accounting for both the intrinsic properties of images and the model's response to resolution changes.

### 3.2.4 EMPIRICAL FORMULA

Intuitively, tasks characterized by high image complexity often necessitate higher input resolutions. Similarly, tasks with high uncertainty variance indicate that increasing resolution heightens model uncertainty, which suggests a need for greater perceptual capacity. Conversely, low uncertainty variance suggests that resolution changes exert minimal impact, making higher resolutions unnecessary. Based on these observations, we hypothesize that image complexity and uncertainty variance are positively correlated with the preferred resolution. Consequently, we propose Equation 3 to determine the optimal resolution for a specific vision-language task $T$:

$$Reso(T) = Reso_0(1 + k \times C(T) \times V(T)) \tag{3}$$

In this empirical formula, $C(T)$ represents the averaged normalized image complexity for task $T$, $V(T)$ denotes the averaged uncertainty variance across different image input resolutions on task $T$, $k$ is a user-specified hyperparameter, and $Reso_0$ is the baseline image input resolution of the original VLLM. The expression $1 + k \times C(T) \times V(T)$ quantifies the scaling factor between the baseline and the preferred resolution. In practice, the value of $k$ can be adjusted based on prior experience.

### 3.3 PARAMETER-EFFICIENT RESOLUTION ADAPTATION

After determining the optimal resolution for a given task, the next step is adapting the VLLM to the selected resolution. To answer **RQ2**, We propose a parameter-efficient fine-tuning (PEFT) approach that post-train an existing VLLM checkpoint, thus avoiding retraining from scratch.

As depicted in Figure 2, existing VLLMs (e.g., LLaVA) consist of three main components: a visual encoder that processes visual inputs, a projector that maps visual features to the word embedding space, and an LLM backbone that autoregressively generates language tokens based on the combined visual and linguistic inputs.

Increasing the input resolution results in more image patches, which introduces incompatibility with the original position embeddings. To address this, we interpolate the position embeddings from the initial number of patches (e.g., $24^2$) to the extended number (e.g., $32^2$), following previous research (Bai et al., 2023; Li et al., 2023b). Although this allows the VLLM to process extended resolutions, performance degrades without further adaptation (as discussed in Secion 3.2). To counter this performance decline, we employ a PEFT method that fine-tunes three key components: (1) position embeddings within the visual encoder, crucial for resolution adaptation due to the change in image patch count; (2) the lightweight projector parameters; and (3) the parameters of the LoRA adapters integrated into the LLM backbone. By keeping all other parameters frozen, the PEFT approach offers an efficient method for adaptation. Figure 2 provides a visual representation of the components that are fine-tuned versus those that remain frozen.

Table 1: A comprehensive investigation conducted to explore resolution preferences across eight vision-language tasks. For each task, the accuracy scores corresponding to five different resolutions are presented.

| Resolution | SciQA-IMG | VizWiz | VQAv2 | GQA | TextVQA | OKVQA | MMBench | MMBench-CN |
|---|---|---|---|---|---|---|---|---|
| $224 \times 224$ | 67.23 | 49.81 | 77.72 | 62.81 | 54.35 | 46.60 | 64.86 | 56.19 |
| $336 \times 336$ | **69.56** | **50.39** | 78.53 | 61.98 | 58.25 | 47.95 | 64.60 | 58.76 |
| $448 \times 448$ | 68.07 | 49.67 | **80.19** | **63.87** | **60.25** | 47.60 | 64.18 | 58.16 |
| $560 \times 560$ | 68.72 | 47.61 | 78.71 | 61.77 | 58.86 | **50.86** | **67.70** | **61.08** |
| $672 \times 672$ | 66.39 | 46.63 | 78.04 | 61.82 | 56.98 | 50.72 | 65.72 | 59.54 |

## 4 EXPERIMENTS

This section presents the empirical evaluation of our proposed method. We first introduce the implementation details in Section 4.1, followed by an in-depth analysis of the results, including the investigation into resolution preferences, task-wise resolution selection, and the findings from the ablation study in Section 4.2, Secion 4.3, and Section 4.4, respectively.

### 4.1 IMPLEMENTATION DETAILS

**VLLM Selection** For our experiments, we select the LLaVA-1.5-7B checkpoint Liu et al. (2023b) as the representative VLLM for evaluation.

**Resolution Configurations** We explore five image resolutions: $224^2$, $336^2$, $448^2$, $560^2$, and $672^2$. These values cover the resolution spectrum commonly used in previous studies Liu et al. (2023b;a).

**Vision-Language Tasks** Our evaluation encompasses eight vision-language tasks, with details introduced in Appendix A.1.

**Baseline Methods** In addition to the original LLaVA model, we compare our method with several state-of-the-art approaches. Besides, we report the performance of position embedding interpolation as a representative of the training-free methods to extend the image input resolution of VLLMs. The details are introduced in Appendix A.2.

**Post-training Details** To initialize the position embedding parameters of the visual encoder (Vision Transformer) in LLaVA during resolution adaptation, we employ extended position embeddings derived through positional embedding interpolation, as described in Appendix A.2. Following the post-training instructions provided by the LLaVA authors[1], we concentrate on stage 2 fine-tuning, incorporating the additional parameters for position embeddings in the visual encoder, alongside the LoRA adapter and projector parameters. The fine-tuning process utilizes images from five datasets: COCO Lin et al. (2014), GQA Hudson & Manning (2019), OCR-VQA Mishra et al. (2019), TextVQA Singh et al. (2019), and Visual Genome Krishna et al. (2017). For more details on the construction of the image-text pairs used in training, we refer readers to Liu et al. (2023a).

More details about method implementation and PEFT are introduced in Appendix A.3 and A.4.

### 4.2 ANALYZING RESOLUTION PREFERENCES ACROSS VISION-LANGUAGE TASKS

We conduct a comprehensive empirical study to analyze the resolution preferences across various vision-language tasks systematically, summarized in Table 1. The findings reveal two key observations:

(1) When image resolution is either too low ($224^2$) or too high ($672^2$), the performance of VLLMs is suboptimal across all tasks. Low-resolution inputs limit the model's ability to capture essential visual details, while very high resolutions create a significant gap between the original and extended image input resolution. This discrepancy leads to less effective resolution adaptation and introduces a greater number of irrelevant image tokens, which do not contribute meaningfully to the specific task at hand.

---

[1]https://github.com/haotian-liu/LLaVA/tree/main?tab=readme-ov-file#train

Table 2: Distributions of image complexity and uncertainty variance accross eight tasks.

| | vizwiz | SciQA-IMG | TextVQA | GQA | VQAv2 | OKVQA | MMBench | MMBench-CN |
|---|---|---|---|---|---|---|---|---|
| Resolution Preference | $336 \times 336$ | | $448 \times 448$ | | | $560 \times 560$ | | |
| Complexity (C) | 0.2191 | 0.1437 | 0.2919 | 0.3236 | 0.3017 | 0.3112 | 0.2323 | 0.2329 |
| Average | | 0.1814 | | 0.3058 | | | 0.2588 | |
| Uncertainty Variance (V) | 1.83% | 6.47% | 4.88% | 5.34% | 5.26% | 6.72% | 10.79% | 10.45% |
| Average | | 4.15% | | 5.16% | | | 9.32% | |
| C × V | 0.0040 | 0.0093 | 0.0142 | 0.0173 | 0.0159 | 0.0209 | 0.0251 | 0.0243 |
| Average | | 0.0067 | | 0.0158 | | | 0.0234 | |

(2) The optimal resolutions are distributed among the intermediate values of $336^2$, $448^2$, and $560^2$, suggesting that the specific visual granularity required by each task varies. No fixed resolution yields optimal performance across all tasks, underscoring the importance of a task-wise resolution selection strategy.

After identifying task-specific resolution preferences, we explore the correlation between optimal resolutions and our proposed heuristics of image complexity and uncertainty variance, as shown in Table 2. Several conclusions can be drawn:

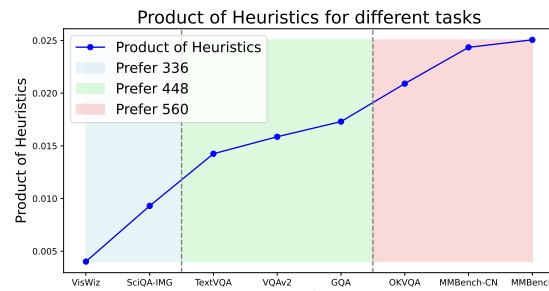

(1) While there is no increasing trend between $448^2$ and $560^2$ in terms of image complexity, a noticeable gap exists between $336^2$ and $448^2$, indicating that image complexity effectively distinguishes tasks preferring $336^2$ from those favoring higher resolutions.

Figure 3: The product of two heuristic scores exhibits a consistent and robust correlation with resolution preferences.

(2) There is a general positive correlation between the preferred resolution and the uncertainty variance of VLLMs across tasks. Averaging the uncertainty variance for each resolution reveals a clear upward trend, indicating that uncertainty variance serves as a reliable indicator for resolution preference.

(3) Although these heuristics generally perform well, some exceptions exist. For instance, GQA prefers a lower resolution than MMbench but has higher image complexity. Similarly, SciQA-IMG exhibits higher uncertainty variance but favors a lower resolution than TextVQA. By combining both heuristics (multiplying their scores), a more consistent and robust correlation is observed, as shown in Figure 3.

### 4.3 Evaluating Heuristic-Based Task-Specific Resolution Selection

The investigation highlights a positive correlation between task-specific resolution preferences and the two heuristic approaches, particularly when their scores are combined. This section introduces

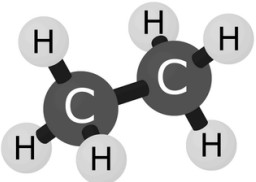

(a) Single and simple object: Ethane is (). A. an elementary substance B. a compound

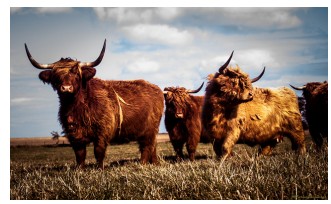

(b) Middle-level complexity: Are all the animals the same?

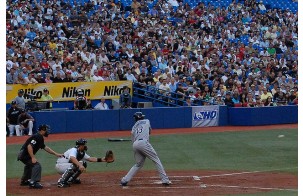

(c) Multiple objects: What is the brand being advertised?

Figure 4: We pick three tasks with images in different levels of complexity.

Table 3: Comparison between our method and baseline approaches, highlighting the best scores in bold. *indicates that the training images or annotations of the datasets were observed during training.

| Method | LLM | Resolution | Post-training | VQAv2 | GQA | TextVQA | OKVQA | MMBench | MMBench-CN |
|---|---|---|---|---|---|---|---|---|---|
| BLIP-2 | Vicuna-13B | $224 \times 224$ | - | 65.00 | 41.00 | 42.50 | - | - | - |
| InstructBLIP | Vicuna-7B | $224 \times 224$ | - | - | 49.20 | 50.10 | - | 36.00 | 23.70 |
| InstructBLIP | Vicuna-13B | $224 \times 224$ | - | - | 49.50 | 50.70 | - | - | - |
| Shikra | Vicuna-13B | $224 \times 224$ | - | $77.40^*$ | - | - | - | 58.80 | - |
| IDEFICS-9B | LLaMA-7B | $224 \times 224$ | - | 50.90 | 38.40 | 25.90 | - | 48.20 | 25.20 |
| IDEFICS-80B | LLaMA-65B | $224 \times 224$ | - | 60.00 | 45.20 | 30.90 | - | 54.50 | 38.10 |
| Qwen-VL | Qwen-7B | $448 \times 448$ | - | $78.80^*$ | $59.30^*$ | $\mathbf{63.80^*}$ | - | 38.20 | 7.40 |
| Qwen-VL-Chat | Qwen-7B | $448 \times 448$ | - | $78.20^*$ | $57.50^*$ | $61.50^*$ | - | 60.60 | 56.70 |
| LLaVA-1.5 | Vicuna-7B | $336 \times 336$ | - | $78.53^*$ | $61.98^*$ | 58.25 | 47.95 | 64.60 | 58.76 |
| LLaVA-1.5 | Vicuna-7B | $448 \times 448$ | ✗ | $77.82^*$ | $61.29^*$ | 56.61 | 47.38 | 63.32 | 57.73 |
| LLaVA-1.5 | Vicuna-7B | $448 \times 448$ | ✓ | $\mathbf{80.19^*}$ | $\mathbf{63.87^*}$ | 60.25 | 47.60 | 64.18 | 58.16 |
| LLaVA-1.5 | Vicuna-7B | $560 \times 560$ | ✓ | $78.71^*$ | $61.77^*$ | 58.86 | $\mathbf{50.86}$ | $\mathbf{67.70}$ | $\mathbf{61.08}$ |
| LLaVA-1.5 | Vicuna-7B | Adaptive | ✓ | $\mathbf{80.19^*}$ | $\mathbf{63.87^*}$ | 60.25 | $\mathbf{50.86}$ | $\mathbf{67.70}$ | $\mathbf{61.08}$ |
| LLaVA-1.5 | Vicuna-13B | $336 \times 336$ | - | $80.00^*$ | $63.30^*$ | 61.30 | - | 67.70 | 63.60 |

the process of determining hyperparameter values in the empirical formula, followed by the overall performance results achieved by models using this selection strategy.

### 4.3.1 APPLYING EMPIRICAL FORMULA TO DETERMINE THE OPTIMAL RESOLUTION

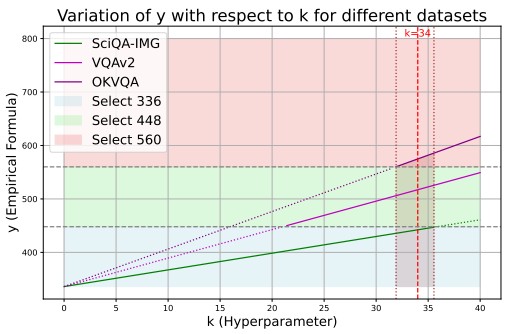 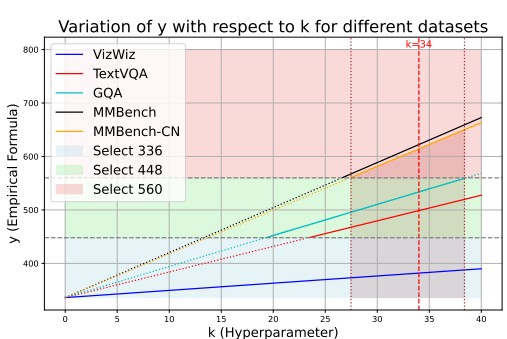

(a) Optimization of the hyperparameters in the empirical formula using three reference tasks.

(b) The empirical formula demonstrates effective generalization across five vision-language tasks.

Figure 5: Applying the empirical formula hyperparameter to determine the optimal resolution for vision-language tasks.

To optimize the hyperparameter in Equation 3, we select three reference tasks that represent varying levels of visual perception requirements for VLLMs. The selection is informed by human experience: tasks associated with simpler images containing a single object (e.g., Figure 4a) are considered as having low demands for image resolution, whereas tasks involving more complex images with multiple objects (e.g., Figure 4c) are associated with higher resolution needs. Intermediate cases (e.g., Figure 4b) are positioned between these extremes. Accordingly, SciQA-IMG, VQAv2, and OKVQA are chosen as representative tasks, reflecting low, medium, and high resolution requirements, respectively.

When tuning the hyperparameter $k$, we focus on the intermediate resolutions $336^2$, $448^2$, and $560^2$. The constant $Reso_0$ in Equation 3 is set to 336 (the default resolution of LLaVA), and the higher resolutions serve as thresholds dividing the empirical formula into three sub-ranges. The task's resolution is determined based on which sub-range its empirical formula value falls into. For example, a value of 500, situated between 448 and 560, results in the selection of resolution $448^2$.

Figure 5a visualizes the relationship between hyperparameter values and selected resolutions. For simplicity, we select $k = 34$, which results in optimal resolution selection for the reference tasks. Additionally, as shown in Figure 5b, this value generalizes well to other tasks, achieving the best resolution for each.

Table 4: Ablation Analysis of PEFT Components, ✗ and ✓ indicate whether the corresponding parameters are post-trained.

| Resolution | ViT PE | Projector | LoRA Adapter | VQAv2 | GQA | TextVQA |
|---|---|---|---|---|---|---|
| $336 \times 336$ | - | - | - | 78.53 | 61.98 | 58.25 |
| $448 \times 448$ | ✗ | ✗ | ✗ | 77.82 | 61.29 | 56.61 |
| $448 \times 448$ | ✓ | ✗ | ✗ | 75.32 | 59.98 | 53.44 |
| $448 \times 448$ | ✗ | ✓ | ✗ | 72.94 | 55.31 | 51.41 |
| $448 \times 448$ | ✗ | ✓ | ✓ | 79.47 | 63.41 | 58.06 |
| $448 \times 448$ | ✓ | ✓ | ✓ | **80.19** | **63.87** | **60.25** |

### 4.3.2 OVERALL RESULTS OF TASK-WISE ADAPTIVE MODEL AND BASELINES

Table 3 presents the performance of baseline methods and LLaVA variants across six tasks that demand high visual perception capacity from VLLMs.

Among the LLaVA variants, the training-free method to extend the input resolution through PE interpolation shows performance degradation at varying levels. This confirms that the position embeddings in the visual encoder and LLM backbone in LLaVA cannot fully adapt to the increased number of image tokens without post-training. On the other hand, the task-wise adaptive LLaVA variant, which optimally selects the input resolution for each task, achieves the best overall performance compared to fixed-resolution LLaVA variants, regardless of whether the resolution is $336^2$, $448^2$, or $560^2$. Notably, the task-wise adaptive LLaVA variant with a 7B backbone performs comparably to the 13B variant, underscoring the importance of adaptive perception capacity in VLLMs.

When comparing the task-wise adaptive LLaVA variant with other state-of-the-art baselines, it outperforms all but the TextVQA task. In the case of TextVQA, the Qwen-VL and Qwen-VL-Chat methods have observed training images or annotations of the dataset during their training. Importantly, as previous studies (McKinzie et al., 2024b) have highlighted, resolution plays a crucial role during pretraining. The Qwen-VL series are pretrained at an image resolution of $448^2$, while the LLaVA variants were fine-tuned at extended image resolutions in a post-training phase with far fewer data (665K) compared to Qwen's 1.4B pretraining and 50M fine-tuning samples. Nevertheless, the task-wise adaptive LLaVA variant achieves better overall results than the Qwen-VL series.

*The superior performance of the task-wise adaptive LLaVA variant across multiple vision-language tasks demonstrates that, compared to fixed-resolution approaches, adaptive resolution selection is more suitable for real-world applications. So far, we have verified the effectiveness of our proposed task-wise resolution selection strategy through the generalization of the empirical formula and the overall experimental results, answering **RQ1**.*

### 4.4 ABLATION ANALYSIS OF PEFT COMPONENTS FOR PERFORMANCE

To evaluate the contribution of each component in our proposed PEFT method, we perform an ablation study, as shown in Table 4. Specifically, we examine the impact of tuning three components of parameters: the position embeddings within the visual encoder, the LoRA adapters in the LLM backbone, and the projector parameters.

The results indicate that tuning each component is crucial for achieving optimal performance. When tuning only the position embeddings or projector parameters, we observe a substantial degradation in performance, even compared to the training-free approach utilizing positional embedding inter-


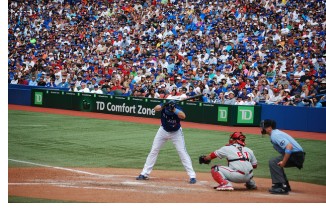
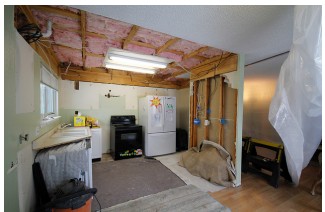

(a)                                           (b)                                           (c)

Figure 6: Three case study images

polation. While the combined tuning of projector parameters and LoRA adapters yields noticeable improvements, the performance remains suboptimal without concurrent tuning of the position embeddings. *These findings highlight the essential role of each component in the PEFT method. So far, through the overall experimental results and the ablation experimental analysis of each component in our proposed PEFT method, we verified the effectiveness of our proposed PEFT method, answering* **RQ2**.

# 5 CASE STUDY

| Image | Figure 6a | Figure 6b |
|---|---|---|
| Question | Who is standing? | |
| Prediction($336 \times 336$) | woman | umpire |
| Correct Answer | woman | batter |
| Image Complexity | 11.35 | 20.62 |

Table 5: Same question with images in different complexity levels.

| Image | Figure 6c | |
|---|---|---|
| Question | What is the sheet made of? | Are there stoves near the freezer to the right of the tap? |
| Prediction($336 \times 336$) | plastic | NO |
| Prediction($448 \times 448$) | plastic | YES |
| Correct Answer | plastic | YES |
| Uncertainty Variance | 0.42% | 16.51% |

Table 6: Same image with questions in different difficulty levels.

In this section, we provide two case studies to illustrate the impact of image complexity and uncertainty variance on the performance of VLLMs, as summarized in Table 5 and Table 6, respectively. The two selected examples are drawn from the GQA dataset (Hudson & Manning, 2019).

Table 5 compares the performance of a VLLM when presented with two images of differing complexity levels, as measured by the method described in Section 3.2.1, both of which are associated with the same question. The question asks the model to identify "who is standing." For the image with lower complexity (Figure 6a), the VLLM at a resolution of $336^2$ correctly identifies the woman standing. Conversely, for the image characterized by a more intricate background (Figure 6b), the model fails to provide the correct identification. This outcome indicates that an increased image input resolution is essential for effectively processing more visually complex images.

Table 6 examines a scenario where the same image is used to answer two questions of differing difficulty. The image shows a room's interior. For the easier question about the material of a sheet, the VLLM at $336^2$ resolution provides a correct answer. However, for the more complex question about the location of a smaller object (a tap), the model fails at $336^2$ but succeeds at $448^2$, highlighting improved performance with higher resolution. Uncertainty variance is low for the simpler question but significantly higher for the complex one, supporting the hypothesis in Section 3.2.4.

# 6 CONCLUSION

In this paper, we take a step towards adapting VLLMs to real-world applications by providing an in-depth investigation of resolution preferences in different vision-language tasks. Based on the findings, we introduce an empirical formula that combines image complexity and uncertainty variance to enable task-specific resolution selection without the need for retraining. Additionally, we propose a parameter-efficient fine-tuning approach, enabling extension of the image input resolution for existing VLLM checkpoints. We expect that our research will offer valuable insights for the VLLM research community.

**Future Work** While this study focuses on LLaVA as a representative VLLM architecture, future work will explore other VLLM architectures. Moreover, our current work centers on task-wise resolution selection; future research will investigate more granular resolution selection strategies, such as sample-level resolution adaptation.

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

## A MORE IMPLEMENTATION DETAILS

### A.1 VISION-LANGUAGE TASKS

*Science-QA* Lu et al. (2022), a multimodal science question answering benchmark featuring over 21k multiple-choice questions on diverse topics. The visual component includes natural images and diagrams, testing the model's ability to integrate both textual and visual information for coherent reasoning and explanation generation. *Vizwiz* Gurari et al. (2018), a dataset derived from real-world images paired with spoken questions from visually impaired individuals. This task assesses a model's ability to process low-quality, unstructured images and generate accurate responses to conversational queries. *VQAv2* Goyal et al. (2017), an expanded version of the original Visual Question Answering (VQA) dataset, designed to reduce language biases. It challenges models to deeply understand visual content in order to answer questions about pairs of semantically similar yet visually distinct images. *TextVQA* Singh et al. (2019), a dataset focusing on a model's capacity to read and reason about textual elements in images, evaluating its ability to integrate Optical Character Recognition (OCR) with visual reasoning to answer questions. *OKVQA* Marino et al. (2019), a benchmark that requires models to leverage external knowledge beyond image and question analysis, necessitating access to and reasoning with unstructured knowledge sources for accurate answers. *GQA* Hudson & Manning (2019), a dataset designed for real-world visual reasoning and compositional question answering, requiring models to demonstrate strong multi-modal understanding, logical reasoning, and the ability to answer questions that necessitate connecting information across both visual and linguistic domains. *MMBench* Liu et al. (2023c), a comprehensive multimodal evaluation set with over 2,974 multiple-choice questions across 20 ability dimensions, providing a robust assessment of various vision-language skills, such as reasoning, comprehension, and explanation generation. *MMBench-CN*, a variant of MMBench focusing on tasks involving Chinese text and images, evaluating the model's proficiency in processing and understanding multilingual data.

### A.2 BASELINE METHODS

In addition to the original LLaVA model, we compare our method with several state-of-the-art approaches, including BLIP-2 Li et al. (2023c), InstructBLIP (Dai et al., 2024) (with LLM backbones at two scales), Shikra (Chen et al., 2023), and IDEFICS (IDEFICS, 2023) (also with LLM backbones at two scales), as well as Qwen-VL and Qwen-VL-Chat Bai et al. (2023). The results for these baseline methods, along with LLaVA with the Vicuna-13B backbone, are cited from previous work (Liu et al., 2023a). For LLaVA with a Vicuna-7B backbone, we report our reproduced results across different vision-language tasks.

As a training-free baseline to extend the image input resolution, we apply positional embedding interpolation to extend the position embeddings of the vision encoder in LLaVA. This technique, widely used for Vision Transformers in VLLMs Bai et al. (2023); Li et al. (2023b), allows models to handle higher image input resolutions than their original training resolution. We evaluate the performance of this extension without any additional training of the projector and the LLM backbone.

Table 7: Hyperparameters at two training stages

| Hyperparameter | batch size | lr | lr schedule | weight decay | epoch | optimizer | max tokens |
|---|---|---|---|---|---|---|---|
| Stage 1 | 256 | 1e-3 | cosinie decay | 0 | 1 | AdamW | 2048 |
| Stage 2 | 128 | 2e-4 | | | | | |

Table 8: Training time cost

| Resolution | $224 \times 224$ | $336 \times 336$ | $448 \times 448$ | $560 \times 560$ | $672 \times 672$ |
|---|---|---|---|---|---|
| Training Time Cost | 11h 50m | 16h 17m | 24h 7m | 32h 29min | 124h 44m |

### A.3 METHOD DETAILS

**Image Complexity Heuristic Approach** Image complexity for vision-language tasks is calculated using an open-source tool[2]. We utilize the author-recommended hyperparameters: the number of clusters is set to 8, and the subsample rate is 0.8. To reduce computational overhead, the input image resolution is set to $112 \times 112$, and two cluster levels are used, with their combined scores yielding the final complexity value. The complexity scores are normalized via min-max scaling, where the minimum and maximum values are computed from 100 sampled images from the ImageNet dataset Deng et al. (2009).

**RandAugment Perturbation on Image Input** When assessing model variance across different resolutions, we apply random perturbations to each input image using the RandAugment algorithm, implemented via an existing tool[3]. For each image, we perform three random augmentations. To mitigate the effects of randomness and enhance result stability, we repeat the variance measurement process three times, each using a different random seed. The final uncertainty variance is obtained by averaging the results from these three iterations.

### A.4 MORE PARAMETER-EFFICIENT FINE-TUNING DETAILS

The standard training hyperparameters are largely preserved, as outlined in Table 7, with two notable adjustments for image resolutions of $560^2$ and $672^2$: (1) The learning rate is reduced from $2e - 5$ to $1e - 5$ to prevent training loss explosion observed with the original rate. (2) The maximum number of tokens is increased from 2048 to 3072 and 4096, respectively, to accommodate the increased number of image tokens.

Post-training experiments are conducted on eight NVIDIA GeForce RTX 4090 GPUs, with training time costs detailed in Table 8. Due to GPU memory limitations, DeepSpeed ZeRO-3 was employed for training at the resolution of $672^2$, while ZeRO-2 was used for other resolutions. This accounts for the significant increase in training time between $672^2$ and $560^2$.

In the ablation study (Section 4.4), we separately fine-tune only the projector and only the position embeddings, using the stage 1 setting for consistency with the goals of the different training stages. The corresponding hyperparameters are also detailed in Table 7.

### A.5 IMPACT OF STATISTICAL DISTRIBUTIONS ON EMPIRICAL FORMULA PERFORMANCE

To evaluate the extent to which the statistical distributions of complexity $C(T)$ and uncertainty variance $V(T)$ influence the performance of the empirical formula, we present the standard deviations of $C(T)$ and $V(T)$ for each vision-language task, along with their respective ratios to the mean values. These statistics are detailed in Table 9.

The results indicate that $C(T)$ exhibits relatively low variance across tasks, whereas $V(T)$ shows substantially higher variability. This observation justifies our decision to adopt task-wise selection

---

[2]https://github.com/Lou1sM/meaningful_image_complexity
[3]https://github.com/TorchSSL/TorchSSL/blob/main/datasets/augmentation/randaugment.py

Table 9: Statistical characteristics of $C(T)$ and $V(T)$ in each task. SD represents Standard Deviation, and Ratio indicates the ratio of the standard deviation to the mean.

| Task | $C(T)$ SD | $C(T)$ Ratio | $V(T)$ SD | $V(T)$ Ratio |
|------|-----------|--------------|-----------|--------------|
| ScienceQA-IMG | 3.3633 | 0.2384 | 0.4398 | 2.5466 |
| Vizwiz | 2.4405 | 0.1541 | 0.3383 | 6.0196 |
| VQAv2 | 2.2005 | 0.1242 | 0.7925 | 4.2562 |
| GQA | 1.6582 | 0.0910 | 1.2595 | 4.9103 |
| TextVQA | 2.3057 | 0.1318 | 0.5258 | 3.3405 |
| OKVQA | 2.1958 | 0.1224 | 0.5487 | 3.7711 |
| MMBench | 3.5426 | 0.2196 | 1.2040 | 2.8915 |
| MMBench-CN | 3.5482 | 0.2197 | 1.0840 | 2.8310 |

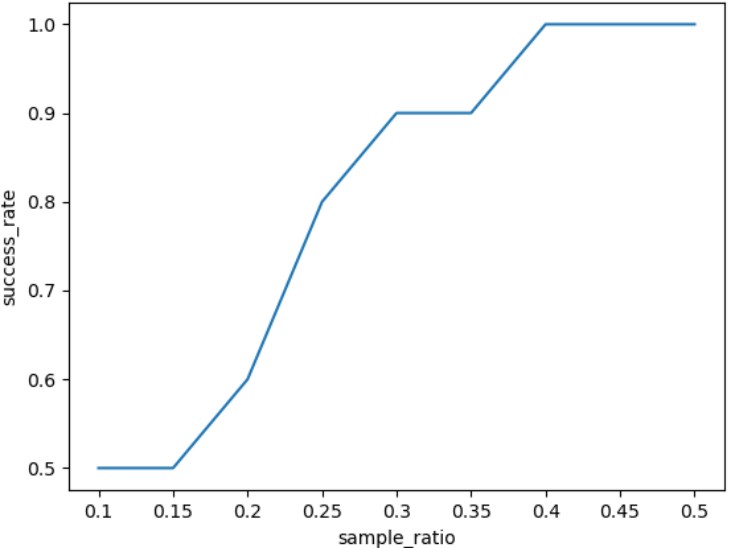

Figure 7: Relationship between sampling ratio and the success rate of the empirical formula.

instead of sample-wise selection, as the higher variability in $V(T)$ at the sample level complicates consistent prediction.

To further assess the influence of $C(T)$ and $V(T)$ variance on the effectiveness of the empirical formula, we conducted an additional experiment. Specifically, we randomly sampled subsets of varying proportions from the original dataset and computed the average $C(T)$ and $V(T)$ values for these subsets to estimate task-level statistics. We then evaluated the empirical formula, previously tuned using a hyperparameter $k$ on three reference tasks, to predict the optimal resolution across all tasks under these conditions.

The sampling proportions vary from 10% to 50%, with each experiment repeated 10 times using different random seeds. The success rate was defined as the percentage of instances where the empirical formula accurately predicted the optimal resolution for all tasks. The results, presented in Figure 7, reveal the following key findings: (1) At a sampling ratio of 40%, the success rate reaches 100%, demonstrating the empirical formula's robustness in predicting the optimal resolution. (2) At a sampling ratio of 10%, the success rate drops to 50%, indicating that a smaller subset size introduces variability that adversely affects prediction accuracy.

These findings highlight that while reducing the dataset size can lower computational costs, excessively small subsets may lead to suboptimal predictions. Moreover, the current approach relies on random sampling; future exploration of more advanced sampling strategies that select representative samples could potentially achieve high success rates with smaller subsets.

