# OpenReview forum: "Beyond Fixed Resolution: Enhancing VLLMs with Adaptive Input Scaling"
_ICLR.cc/2025/Conference — ICLR 2025 Conference Withdrawn Submission_

### Official Review · Reviewer_KqHt · 2024-10-30

**Soundness:** 3
**Presentation:** 3
**Contribution:** 2
**Rating:** 5
**Confidence:** 5

**Summary:**

This paper proposes a "two-stage" adaptive resolution pipeline. The authors introduce an empirical formula for choosing an optimal resolution. Experiments show that this method is effective on LLAVA.

**Strengths:**

The method lifts the performance of LLaVA. The proposed task-driven dynamic resolution is meaningful.

**Weaknesses:**

1. The pipeline is not end-to-end and looks unsightly. It seems like two VLMs cascaded.
2. The baseline, only LLaVA, is limited.
3. I feel it would be best for the authors to design the dynamic resolution of the task-driven mode in an end-to-end manner. The current approach is too heavy in image preprocessing

**Questions:**

See weakness.

---

> ### Author Response · Authors · 2024-11-23
> **Response for Reviewer KqHt**
>
> We sincerely thank Reviewer KqHt for recognizing the significance of our task-driven dynamic resolution approach. Below, we address the reviewer’s concerns point by point.
>
> **[Cons 1. The pipeline is not end-to-end.]**
> We respectfully clarify that the two-stage design of our pipeline is both **necessary** and **natural** for the proposed approach. The "two-stage" nature, as noted by the reviewer, refers to the optimal resolution selection stage and the resolution adaptation stage. However, we argue that these two stages are integral and do not reflect a "cascaded" VLLM design.
>
> **Necessity of Two Stages**
> - The first stage identifies the optimal resolution for a specific task by measuring image complexity and leveraging uncertainty variance scores derived from inference on reference VLLMs.
> - The second stage adapts the VLLM to the chosen resolution using post-training. As highlighted in Section 4.4, training-free approaches (e.g., interpolating ViT’s position embeddings) result in performance degradation, making post-training essential.
>
> These stages complement one another: the first determines the resolution, while the second adapts the model. Importantly, they operate independently and **do not cascade**. The first stage involves inference, whereas the second stage involves post-training.
>
> **Cost Considerations**
>
> Our approach is designed to be efficient:
> - Both stages can be conducted offline, with the resolution adaptation stage employing a parameter-efficient fine-tuning strategy to minimize computational overhead.
> - Once the VLLM is adapted, it operates at a single, optimal resolution during inference, eliminating concerns about heavy preprocessing or the need for multiple VLLMs in production.
>
> This ensures that our pipeline is both resource-efficient and practical for real-world applications.
>
> **[Cons 2. Only LLaVA is used as the baseline.]** We respectfully argue that while our experiments are conducted on LLaVA-1.5, our method is not specifically designed for this model and can be extended to other VLLMs.
>
> **Representativeness of LLaVA**: LLaVA is widely regarded as a representative and prevailing model for multimodal large language model research [1,2], as evidenced by its adoption in numerous recent studies as the **sole** MLLM instance [3,4,5,6]. These studies demonstrate that insights and improvements derived from LLaVA are broadly applicable to other VLLMs.
>
> **Generalizability of our approach**: Our resolution preference investigation spans eight vision-language tasks, with the proposed empirical formula validated across five tasks. These results indicate that our approach generalizes well beyond a single benchmark.
> While we acknowledge that experiments on additional VLLMs would strengthen this claim, the current scope focuses on validating the feasibility of our method. We plan to extend these evaluations in future work to further demonstrate the generalizability.
>
> We hope the above clarifications address the reviewer’s concerns and demonstrate the rationale and practicality of our approach. Thank you again for your constructive feedback.
>
> [1] A Survey on Multimodal Large Language Models \
> [2] A Survey of Large Language Models \
> [3] Video-LLaVA: Learning United Visual Representation by Alignment Before Projection, EMNLP 2024 \
> [4] Honeybee: Locality-Enhanced Projector for Multimodal LLM, CVPR 2024 \
> [5] See, Say, and Segment: Teaching LMMs to Overcome False Premises, CVPR 2024 \
> [6] LITA: Language Instructed Temporal-Localization Assistant, ECCV 2024

---

### Official Review · Reviewer_fQzB · 2024-11-01

**Soundness:** 2
**Presentation:** 2
**Contribution:** 2
**Rating:** 5
**Confidence:** 4

**Summary:**

Regarding the subpar performance of VLLM in downstream tasks when using fixed resolution for image processing, this work proposes a task-wise resolution selection method and adapts the model to optimal resolution on each task through post-training. By measuring image complexity and model uncertainty variance across resolutions and combining the two, an empirical formula for calculating optimal resolution based on the baseline resolution is obtained. In the resolution adaptation process, by adding LORA and training only certain key parameters, the performance of the proposed 7B model in downstream tasks surpasses all baselines and is comparable to the 13B model.

**Strengths:**

1. The overall logic is sound, and the experimental results demonstrate that the method proposed in the paper alleviates the shortcomings of fix resolution in downstream VQA tasks.
2. The article provides comprehensive experiments and analysis on the calculation and validation of optimal resolution.

**Weaknesses:**

1. The research findings of the article are relatively superficial, lacking deeper exploration. The relationship between the task and its optimal resolution in this paper seems more like overfitting to the benchmark itself. It is more important to focus on the sample-wise resolution selection or to derive the relationship between optimal resolution and more abstract task categories, rather than specific benchmarks.
2. There is a lack of comparison with other dynamic resolution methods, such as the dynamic number of tiles used in InternVL[1].
3. There are too few captions for the figures and tables. To understand the details of the figures and tables, one needs to refer back to a specific section.

[1] Chen, Zhe, et al. "Internvl: Scaling up vision foundation models and aligning for generic visual-linguistic tasks." Proceedings of the IEEE/CVF Conference on Computer Vision and Pattern Recognition. 2024.

**Questions:**

1. Intuitively, the selection of optimal resolution should not only be related to image complexity and uncertainty variance but also to the specific QA. For example, asking whether a photo contains a panda versus asking about the number of bamboo in a panda's arms would require different levels of image detail, which is more crucial than image complexity itself. The paper lacks sufficient analysis on this aspect.
2. According to Eq.3 in the paper, the task-wise optimal resolution is determined by C(T) and V(T), where C(T) and V(T) are the means of all samples in the task. What are the statistical distributions of these two values in each task? If there is significant variance, would it affect the significance of the mean itself?
3. The article does not provide specific information about the training data. It would be helpful to include more details about the experiments in the experimental section. Conversely, the case study section seems a bit lengthy.

---

> ### Author Response · Authors · 2024-11-24
> **Response for Reviewer fQzB [Cons 1-3]**
>
> We sincerely thank Reviewer fQzB for acknowledging that “the overall logic is sound,” our method “alleviates the shortcomings of fixed resolution in downstream VQA tasks,” and that our experiments are “comprehensive.” Below, we provide responses to the reviewer’s concerns and questions point by point.
>
> **[Cons 1. Findings of the article are relatively superficial.]**
> We respectfully clarify that the primary contribution of our work lies not in **the empirical formula itself** but in uncovering the **underlying relationship between image complexity, uncertainty variance, and optimal resolution**. These insights offer a broader understanding of the factors influencing task-specific resolution preferences. Furthermore, the empirical formula has been validated across five vision-language tasks, demonstrating its generalizability rather than overfitting to specific benchmarks.
>
> Regarding to the sample-level resolution selection, we agree with the reviewer that it is an intriguing and valuable direction. **However, it is also significantly more challenging**. Our preliminary explorations revealed that:
> - **High Variance Across Samples**: Individual samples exhibit substantial variability especially in uncertainty, making consistent predictions for optimal resolution challenging (Please kindly refer to **our response to Q2** on statistical distributions for more details).
> - **Challenges in Predictor Developing at Sample Level**: We found it difficult to develop an effective predictor to determine the optimal resolution on sample level. Due to the high variance across samples, the predictor cannot effectively transfer from training data to test data. Similar difficulties have been identified in a recent work [1], where researchers observed a large gap between the total number of visual tokens and the number of truly informative tokens. They also struggled to develop an accurate predictor for this "oracle" number of tokens at the sample level.
>
> Given these challenges, we opted for task-level resolution selection as a more **viable** and **practical** approach, allowing us to reliably verify the relationships among image complexity, uncertainty variance, and optimal resolution.
>
> As suggested, we plan to expand on sample-level resolution selection in future work, as highlighted in the Future Work section of our paper.
>
> Additionally, to address the reviewer’s concern about the depth of our analysis, **we have included the statistical distribution of image complexity and uncertainty variance for each task in the revised version, offering a more granular view of the data**.
>
> [1] Matryoshka Multimodal Models
>
> **[Cons 2. Lack of comparison with other dynamic resolution methods.]**
> We sincerely thank the reviewer for suggesting a comparison with other dynamic resolution methods. However, we would like to clarify that the mentioned work, InternVL, **does not support dynamic resolution** as claimed. While this model involves multiple resolutions during its Stage-1 training (initially trained at $196^2$ and later switched to $224^2$), it uses **a fixed input resolution during inference**. In contrast, our approach determines the optimal resolution for each vision-language task and adapts the model to these resolutions. Crucially, our method supports the use of **varying resolutions for different vision-language tasks** during inference, which is a fundamental difference from InternVL.
>
> Additionally, we acknowledge that there are other works supporting dynamic resolutions at the inference stage, such as Qwen2VL and MiniCPM-V2, as noted by Reviewer bpfC. However, our approach fundamentally differs from these methods in several aspects, as detailed in our response to Cons 1 raised by Reviewer bpfC.
>
> **[Cons 3. too few captions for the figures and tables]**
> We greatly appreciate this valuable feedback. **We have thoroughly revised the captions for all figures and tables to make them more descriptive**. These improvements, **highlighted in red in the revised PDF**, ensure that readers can understand the key visual elements without needing to reference specific sections of the text.

---

> ### Author Response · Authors · 2024-11-24
> **Response for Reviewer fQzB [Q1&Q2]**
>
> **[Q1. selection of optimal resolution should be related to the specific QA]**
> We agree that the selection process should consider the specific question-answer (QA) pair. **In fact, this consideration is integral to our method and is why we incorporate the uncertainty variance component**. Specifically, in addition to image complexity, our method measures the difference in logit entropy generated by two reference VLLMs. Since the entropy is derived from the output logits of the LLM backbone (i.e., the final layer), it **inherently reflects the interaction between visual and textual token representations** during the forward pass. This ensures that the generated uncertainty is consistent with the specific QA pair under consideration.
> This motivation is explained in Section 3.2.2 of our paper, where we state:
>
> >“...as it provides insights into the interaction between the visual and linguistic components of vision-language tasks.”
>
> >“This heuristic approach serves two functions: (1) it computes entropy based on the tokens generated by VLLM, thus accounting for both visual and linguistic features during inference…”
>
> Therefore, in scenarios like the reviewer’s example—where different questions are asked about the same image (e.g., identifying a panda versus counting bamboo in its arms)—the varying demands for image details are effectively captured through the uncertainty variance.
>
> Moreover, we demonstrate **a similar case in the case study section (Section 5)**. For the same image, different questions produce varying levels of uncertainty variance, aligning with the differing detail requirements of the questions. This supports the robustness of our method in addressing specific QA demands.
>
>
> **[Q2. What are the statistical distributions of C(T) and V(T) in each task? If there is significant variance, would it affect the significance of the mean itself?]**
>
>
> To address this question, we provide the standard deviation of C(T) (image complexity) and V(T) (uncertainty variance) for each vision-language task, along with their respective ratios to the mean values, as shown in **Table R1**:
>
>
> **Table R1.**  Statistical distributions of C(T) and V(T) in each task. SD indicates Standard Deviation, and Ratio indicates the ratio of Standard Deviation to the mean.
> | Task          | C(T) SD | C(T) Ratio | V(T) SD | V(T) Ratio |
> |---------------|---------|------------|---------|------------|
> | ScienceQA-IMG | 3.3633  | 0.2384     | 0.4398  | 2.5466     |
> | Vizwiz        | 2.4405  | 0.1541     | 0.3383  | 6.0196     |
> | VQAv2         | 2.2005  | 0.1242     | 0.7925  | 4.2562     |
> | GQA           | 1.6582  | 0.0910     | 1.2595  | 4.9103     |
> | TextVQA       | 2.3057  | 0.1318     | 0.5258  | 3.3405     |
> | OKVQA         | 2.1958  | 0.1224     | 0.5487  | 3.7711     |
> | MMBench       | 3.5426  | 0.2196     | 1.2040   | 2.8915     |
> | MMBench-CN    | 3.5482  | 0.2197     | 1.0840   | 2.8310     |
>
> From these results, we observe that C(T) exhibits relatively low variance across tasks, while V(T) shows higher variability. This discrepancy reinforces our decision to perform task-level rather than sample-level resolution selection, as the higher variance in V(T) at the sample level makes consistent predictions more challenging.
>
> To evaluate whether the variance of C(T) and V(T) impacts the robustness of our method, we conducted an additional experiment involving random sampling. Specifically, we sampled subsets of the dataset at various ratios and used the mean values of C(T) and V(T) from these subsets to estimate task-level parameters. The empirical formula (with hyper-parameter k tuned on three reference tasks) was then used to predict the optimal resolution for all tasks.
>
> The results of this sampling experiment, **detailed in Figure 7 (in Appendix A.5 of the revised paper)**, are summarized as follows:
> - At a 40% sampling ratio, the success rate of predicting the optimal resolution reaches 100%, demonstrating the robustness of our empirical formula under sufficient sampling conditions.
> - At a 10% sampling ratio, the success rate drops to 50%, indicating that insufficient sampling can lead to inaccuracies due to the higher variance in sample distributions.
>
> These findings suggest that while computation costs can be reduced by using smaller subsets, ensuring sufficient sampling is critical for reliable predictions. Moreover, we will explore whether more sophisticated sampling strategies could potentially achieve accurate predictions at lower sampling ratios in the future work.
>
> We sincerely thank the reviewer for raising this insightful question. **The discussion and results have been incorporated into Appendix A.5 of the revised paper**. We believe these additions enhance the understanding of our method’s robustness and practical applicability.

---

> ### Author Response · Authors · 2024-11-24
> **Response for Reviewer fQzB [Q3]**
>
> **[Q3: Lack of training data details and adjustment of section lengths.]**
> Details about the training data are provided in Appendix A.4, where we clarify that our post-training follows LLaVA’s Stage 2 of training process. This includes a reference to the official GitHub repository, which specifies the datasets and hyperparameters used. Since we strictly adhere to LLaVA’s training settings, we initially directed readers to this source. However, based on the reviewer’s feedback, we have included additional details in the revised paper for clarity and completeness.
>
> Regarding the adjustment of section lengths, we have shortened the case study section to be more concise while expanding the experiments section to provide additional implementation details. We believe these revisions will create a more balanced and informative presentation of our work.
>
> **Please kindly refer to the corresponding content in Section 4.1 and Section 4.5 (the extended content in Section 4.1 is color in red).**
>
> We hope these clarifications address the reviewer’s concerns and highlight the rationale and contributions of our approach. Thank you again for your constructive feedback.

---

> ### Comment · Reviewer_fQzB · 2024-11-28
> **Response to Authors**
>
> I appreciate the author's detailed responses to my questions and concerns.
>
> 1. I no longer have further questions regarding the specifics of the training data.
> 2. I now clearly see the difference between this paper's approach and the VLLMs supporting dynamic resolution.
> 3. The author has provided a reasonable explanation regarding my primary concern, which is the consideration of question text in this paper's method.
> 4. While I was hoping for a sample-wise resolution selection result for the main parameters C(T) and V(T) in the method, the high variance of V(T) primarily stems from the instability of the baseline VLLM(or other VLLMs) itself, which should not be considered a flaw of this paper's method.
>
> The method in this paper is logically sound. Although I still believe that the contribution of this paper's method is not as significant as that of VLLMs supporting dynamic resolution, I maintain my judgment. Still, I am willing to raise the score to 5.

---

### Official Review · Reviewer_bpfC · 2024-11-02

**Soundness:** 2
**Presentation:** 1
**Contribution:** 2
**Rating:** 5
**Confidence:** 5

**Summary:**

This paper investigates the resolution preferences of different vision-language tasks and proposes an empirical formula to determine optimal resolution. It also presents a novel parameter-efficient fine-tuning technique to extend the visual input resolution of pre-trained models, which is validated by extensive experiments.

**Strengths:**

This paper explores the resolution preferences of different vision-language tasks and formulates an empirical formula to determine a relatively appropriate resolution. It also proposes a new parameter-efficient fine-tuning technique to enhance the visual input resolution of pre-trained models.

**Weaknesses:**

1. The author's starting point is good. However, the assumption that all existing VLLMs have a fixed resolution is invalid. Many existing VLLMs are of dynamic resolution, such as MiniCPM-V2 and Qwen2VL. The author seems to have directly ignored such methods and there is no discussion on them at all.

2. Regarding task selection, I don't seem to see that the author has selected tasks that are highly dependent on resolution for statistical evaluation, such as the DocVQA dataset. From my experience, this is a task scenario that is highly dependent on resolution. Statistical data in this task scenario can provide some inspiration.

**Questions:**

See the weakness.

---

> ### Author Response · Authors · 2024-11-24
> **Response for Reviewer bpfC**
>
> We sincerely thank Reviewer bpfC for recognizing the novelty of our proposed parameter-efficient fine-tuning technique and the comprehensiveness of our experimental validation. Below, we address the reviewer’s concerns point by point.
>
> **[Cons 1: Assumption that all existing VLLMs have a fixed resolution is invalid.]**
> We respectfully clarify that we do not claim **all** existing VLLMs have a fixed resolution. Instead, as stated in our paper, **most** existing VLLMs pre-assume a fixed resolution for downstream tasks, which remains a valid assertion supported by prior work [1,2,3].
> Regarding the mentioned MiniCPM-V2 and Qwen2VL, we agree that these models can handle dynamic resolutions. However, our approach and theirs differ fundamentally in the following ways:
>
> - **Design Objectives**:
> Our work focuses on designing an efficient and lightweight post-training schema to adapt existing VLLM checkpoints to new resolutions. In contrast, MiniCPM-V2 and Qwen2VL are designed for training from scratch, necessitating more extensive computational resources.
> - **Dynamic Resolution Mechanisms**:
> We emphasize adapting resolution preferences based on **the varying perceptual demands of different vision-language tasks**. Conversely, MiniCPM-V2 references [4] and achieves dynamic resolution through a partition-based strategy. For Qwen2VL, it modifies the position embedding schema of ViT to incorporate 2D-RoPE. They focus **solely on architectural adjustments** for dynamic resolution and do not account for task-specific resolution requirements, which may be less efficient for some cases which have a lower demand of perception capacity.
> - **Incorporating Textual Factors**:
> Unlike these works, which consider resolution **solely from the image perspective**, our method integrates image complexity and uncertainty variance, which account for **both visual and textual aspects**. This dual consideration reflects the interaction between image and language modalities, providing a more holistic adaptation framework.
>
> These differences highlight that while MiniCPM-V2 and Qwen2VL address dynamic resolution, their objectives and approaches diverge significantly from ours. Still, we will add a discussion in the related work section to explicitly acknowledge these methods and clarify their distinctions, if the reviewer finds it necessary.
>
> **[Cons 2. selection of tasks that are highly dependent on resolution]**
> We appreciate the reviewer’s insightful suggestion to evaluate resolution-dependent tasks. While our current evaluation spans eight diverse vision-language tasks that cover a range of resolution dependencies, we recognize the value of including document-centric datasets like DocVQA to further enrich the statistical analysis.
> As part of our future work, we plan to extend our evaluation to include such resolution-sensitive tasks and explicitly compare resolution-dependent tasks (e.g., DocVQA) with resolution-agnostic tasks. This will provide a more nuanced understanding of how task-specific characteristics influence resolution preferences and further validate the robustness of our proposed approach.
>
> [1] Chen K, Zhang Z, Zeng W, et al. Shikra: Unleashing multimodal llm’s referential dialogue magic. arXiv preprint arXiv:2306.15195, 2023. \
> [2] Dai W, Li J, Li D, et al. InstructBLIP: Towards general-purpose vision-language models with instruction tuning. arXiv preprint arXiv:2305.06500, 2023. \
> [3] Liu H, Li C, Wu Q, et al. Visual instruction tuning. NeurIPS, 36, 2024. \
> [4] Guo Z, Xu R, Yao Y, et al. LLaVA-UHD: an LMM Perceiving Any Aspect Ratio and High-Resolution Images. ECCV, 2025.

---

> > ### Comment · Reviewer_bpfC · 2024-12-02
> >
> > I appreciate the author's detailed responses to my questions and concerns.
> >
> > Regarding the first issue, I understand the author's explanation.
> > However, concerning the second point, the author has merely outlined such a plan without implementation. I maintain my position that the author did not select evaluation tasks that depend on high-resolution processing, a limitation which the author has acknowledged. Therefore, I believe the manuscript requires further development and refinement. I maintain my original rating score.

---

### Official Review · Reviewer_RKQF · 2024-11-02

**Soundness:** 3
**Presentation:** 3
**Contribution:** 3
**Rating:** 6
**Confidence:** 4

**Summary:**

This paper presents an investigation of the optimal image resolution for different vision-language tasks. It then propose a parameter-efficient fine-tuning technique to extend pretrained VLLMs to the target resolution.

**Strengths:**

This paper presents an interesting investigation in the choice of image resolution for different vision-language tasks with VLLMs. It reveals that image resolution would influence the performance for different downstream tasks. To solve the resolution variants problem, the authors futher propose a parameter-efficient fine-tuning techinque to tailor pretrained VLLMs to different image resolution. The experiments are extensive and well-organized.

**Weaknesses:**

Investigation on only LLaVA models might impair the generalization of this analysis.

**Questions:**

Does discrete image representation in VLLMs also suffer from the same image resolution problems?

---

> ### Author Response · Authors · 2024-11-23
> **Response for Reviewer RKQF**
>
> We sincerely thank Reviewer RKQF for recognizing our investigation as “interesting” and describing our experiments as “extensive and well-organized.” Below, we address the reviewer’s concerns and questions point by point.
>
> **[Cons: Investigation on LLaVA only might impair generalization.]**
> We respectfully argue that our method is not specifically designed for LLaVA-1.5, and it can be extended to other VLLMs. LLaVA-1.5 was chosen as a representative model due to its prevalence and importance in multimodal large language model research. Many existing VLLMs are built on the LLaVA framework, as evidenced by works such as [1,2].
> Additionally, many recent publications use LLaVA as the sole VLLM instance for analysis, demonstrating its representativeness in the field. For instance, [3,4,5,6] all adopt LLaVA as the **sole** VLLM instance for architecture design or evaluation. These examples indicate that insights and improvements on LLaVA are broadly applicable to VLLMs in general.
> Finally, the resolution preference investigation covers eight vision-language tasks, and the proposed empirical formula remains valid across five tasks, further verifying the generalization of our method.
>
> **[Question: Does discrete image representation in VLLMs also suffer from the same image resolution problems?]**
> Yes. The LLM backbone in VLLM also suffers from this problem, since image input resolution changes directly affect the number of visual tokens (i.e. discrete image representation) processed by the LLM backbone. For instance, increasing the image input resolution from $336^2$ to $448^2$ results in an increase in the visual token counts from $24^2$ to $32^2$. While LLMs can handle variable-length input, prior work [7] suggests that tokens in middle positions may be less effectively utilized, potentially degrading performance.
> Moreover, as LLaVA is pre-trained at a fixed resolution of $336^2$, its LLM backbone may not adapt well to changes in token counts caused by resolution shifts. To address this, we employ LoRA technique to post-train the LLM backbone, enabling it to adapt to the increased visual tokens.
> Empirical validation of this observation is provided in Section 4.4 of our paper. Ablation studies show that freezing the LLM backbone and only tuning the visual encoder leads to performance degradation. In contrast, post-training the LoRA adapters significantly improve performance, highlighting the necessity of tuning the LLM backbone parameters.
>
> [1] A Survey on Multimodal Large Language Models \
> [2] A Survey of Large Language Models \
> [3] Video-LLaVA: Learning United Visual Representation by Alignment Before Projection, EMNLP 2024 \
> [4] Honeybee: Locality-Enhanced Projector for Multimodal LLM, CVPR 2024 \
> [5] See, Say, and Segment: Teaching LMMs to Overcome False Premises, CVPR 2024 \
> [6] LITA: Language Instructed Temporal-Localization Assistant, ECCV 2024 \
> [7] Lost in the Middle: How Language Models Use Long Contexts, TACL 2024

---

> > ### Comment · Reviewer_RKQF · 2024-12-02
> >
> > Thanks for the detailed response which addresses my concern. I don't have any other questions and I would like to maintain my rating for a weak acceptance.

---

### Note · Authors · 2024-12-15

**Comment:**

We sincerely appreciate the time and effort the reviewers have dedicated to evaluating our submission, as well as the valuable and insightful feedback provided. After careful consideration, we have decided to withdraw this submission to further improve it based on the reviewers' constructive suggestions.

**Withdrawal Confirmation:**

I have read and agree with the venue's withdrawal policy on behalf of myself and my co-authors.